# Deep Reinforcement Learning Based Resource Allocation with Radio Remote Head Grouping and Vehicle Clustering in 5G Vehicular Networks

Hyebin Park  and Yujin Lim *

Department of IT Engineering, Sookmyung Women's University, Seoul 04310, Korea; hb0390@sookmyung.ac.kr
* Correspondence: yujin91@sookmyung.ac.kr; Tel.: +82-2-2077-7305

**Abstract:** With increasing data traffic requirements in vehicular networks, vehicle-to-everything (V2X) communication has become imperative in improving road safety to guarantee reliable and low latency services. However, V2X communication is highly affected by interference when changing channel states in a high mobility environment in vehicular networks. For optimal interference management in high mobility environments, it is necessary to apply deep reinforcement learning (DRL) to allocate communication resources. In addition, to improve system capacity and reduce system energy consumption from the traffic overheads of periodic messages, a vehicle clustering technique is required. In this paper, a DRL based resource allocation method is proposed with remote radio head grouping and vehicle clustering to maximize system energy efficiency while considering quality of service and reliability. The proposed algorithm is compared with three existing algorithms in terms of performance through simulations, in each case outperforming the existing algorithms in terms of average signal to interference noise ratio, achievable data rate, and system energy efficiency.

**Keywords:** vehicular networks; V2X communication; deep reinforcement learning; vehicle clustering



## 1. Introduction

With the increasing data traffic in wireless and vehicular networks, it has become difficult to meet data traffic requirements. To this end, cellular network architectures have been researched as a potential solution. In traditional architectures, one base station (BS) is deployed in one cell to provide voice and data services. To improve the bandwidth for 5G cellular networks, many architectures have been studied including heterogeneous network (HetNet), cloud radio access network (C-RAN), and heterogeneous cloud radio access network (H-CRAN).

In HetNet, one macro-cell and multi-small cells are overlapped to satisfy the rapidly growing data traffic requirements. The small cells cover a smaller range than the macro cell and are deployed close to users to provide a higher data rate. However, small cells also consume more system energy at BSs than traditional network architectures. To minimize the system energy consumption, the BS structure is divided into baseband unit (BBU) and remote radio head (RRH) in C-RAN. The BBUs that process signals are centralized in a BBU pool and RRHs that serve as radio-frequency transceivers are distributed in the cells. However, it also causes serious interference because of the dense deployment. The H-CRAN has been developed to utilize the advantage of both HetNet and C-RAN [1]. In one cell of H-CRAN, multiple small RRHs are connected to the BBU pool through fronthaul links and a macro BS is connected to the BBU pool through the backhaul link. The macro BS functions only as a voice service, and RRHs function only as wireless data services. Hence, the BBU pool can manage and allocate communication resources efficiently.

Device-to-device (D2D) communication has also been developed to satisfy the high data rate requirements. With D2D communication, nearby wireless devices can communicate directly without RRHs, reducing the system energy consumption. However, the

rapidly varying channel condition owing to high mobility in the vehicular environment makes it difficult to collect instantaneous channel state information (CSI) at RRH. Unlike common wireless networks, data traffic in vehicular networks can have regularities in the spatial—temporal dimension as a result of the periodicity in urban traffic flow [2]. Hence, considerable research has been conducted on vehicle-to-everything (V2X) communication to improve road safety and to ensure reliable and low latency services [3–6]. Two communication modes in cellular V2X, vehicle-to-infrastructure (V2I) and vehicle-to-vehicle (V2V) have been modified to accommodate various vehicular applications.

In this paper, we propose a deep reinforcement learning (DRL) based resource allocation algorithm with a clustering method in cellular V2X communication. In our algorithm, neighboring RRHs are grouped to reduce signaling overhead in DRL deployments. Vehicles are clustered to collect periodical messages and request resource allocation to minimize system energy consumption. The DRL based resource allocation learns an optimal allocation strategy to maximize the sum of achievable data rates. As there is a trade-off relationship between the energy consumption and achievable data rate, it is important to identify the strategy that optimizes both objectives. Hence, the main goal of our proposed algorithm is to maximize the system energy efficiency while ensuring quality of service (QoS) and reliability.

The remainder of this paper is organized as follows. Section 2 provides an overview of related works. In Section 3, we describe the system model and formulate the problem. Section 4 presents the proposed algorithm, which consists of RRH grouping, vehicle clustering, and DRL based resource allocation. A performance evaluation and discussion on our proposed algorithm are presented in Section 5. Finally, we present the conclusion in Section 6.

## 2. Related Works

Several optimization methods have been studied to improve system energy efficiency and QoS in vehicular networks [7]. In ref. [8], a context based scheduling scheme was proposed that uses the geographical location of vehicles and the interference level of shared resources. The distribution of interference is obtained through resource allocation and the reduction of packet collisions ensures stable management of resources. The cluster method classifies vehicles into different groups to facilitate dissemination of messages to other vehicles, as a typical method of vehicular communication. In ref. [9], a low-cost cluster-based communication method, which includes data estimation, was developed to improve the QoS of driver assistance with crash warning application. The intra-cluster communication method was applied for data estimation to reduce the additional cost of inter-cluster communication.

It is difficult to ensure QoS requirements of the various existing algorithms that exploit traditional optimization methods. Machine learning has shown promise in addressing decision-making problems involving uncertain network communications [10]. The method finds optimal or close to optimal solutions in the uncertain and inherently non-stationary environment of vehicular networks. In [11], a reinforcement learning scheme was developed to fulfill the diverse requirements of different entities with stringent QoS requirements. This algorithm can minimize the overhead of resource provisioning for vehicular clouds. In [12], an online reinforcement learning based user association algorithm for network balancing was proposed to cope with dynamic changes using historical-based reinforcement learning. In [13], a two-level cluster-based routing method was developed for efficient data dissemination by adjusting two cluster methods, level-1 cluster heads and level-2 cluster heads. The level-1 cluster heads were selected by applying fuzzy logic to relative velocity, k-connectivity and link reliability factors. The level-2 cluster heads were selected by Q-learning to reduce communication overhead.

In vehicular networks with multiple sensing components and realistic channel gain, high mobility generates large-scale continuous state space, which makes Q-learning inefficient. DRL, which uses deep neural networks (DNN) to represent the Q-table, can be

modified to solve resource allocation problem in vehicular networks such that the continuous state is a direct input to the DNN [14]. In [15], the deep deterministic policy gradient (DDPG) based resource allocation scheme was proposed to maximize the sum rate of V2I communications, where the resource allocation problem is formulated as a decentralized discrete-time and finite-state Markov decision process (MDP) with incomplete network information. DDPG is capable of handling continuous high-dimensional action spaces while determining the optimal resource allocation strategy. In [16], a decentralized resource allocation mechanism based on DRL was proposed to maximize the achievable data rate and minimize the interference of V2I communications. DRL can be modified to both unicast and broadcast scenarios with a V2V vehicle as an agent. The agent decides on the optimal sub-band and transmission power levels. In [17], a DRL based decentralized mode selection and resource allocation algorithm was proposed to maximize the sum capacity while meeting the latency and reliability requirements. The mode selection and resource allocation problems were formulated as a Markov decision process to select the optimal transmission mode, sub-band and transmission power level. To address the limitations of local DRL models, a two-timescale federated DRL and graph theory based vehicle clustering method was developed.

High mobility in vehicular networks results in frequent handover and channel state changes. The aforementioned studies consider neither realistic vehicle mobility nor changing channel states. Channel state changes caused by high mobility result in the frequent update of channel information, causing signaling overhead and latency. Therefore, an appropriate method is required to maximize system energy efficiency while reducing signaling overhead and guaranteeing latency requirements. Typical optimization methods are highly complex and incur signaling overhead. The Q-learning method is not capable of processing high-dimensional data or learning directly the network data. Hence, there is need for research on a DRL based technique capable of reducing the signaling overhead of channel state updates generated from vehicle mobility.

In this paper, we present a DRL based resource allocation algorithm with RRH grouping and vehicle clustering for cellular V2X communication while guaranteeing QoS and reliability requirements. Our goal is to maximize system energy efficiency. The proposed algorithm consists of three parts: One for RRH grouping, for vehicle clustering, and a DRL based resource allocation method. The RRH grouping method is a clustering technique that groups neighboring RRHs to reduce the communication overhead and maintenance cost of DRL. The vehicle clustering method clusters vehicles to reduce the signaling overhead of periodic messages. Each vehicle cluster constructs a jumbo frame from the periodic messages of cluster members and requests resource allocation. To allocate communication resources, the DRL based resource allocation is adjusted. Agents are deployed to each RRH, to learn to maximize the sum capacity of the system so that the RRH grouping and vehicle clustering method can minimize system energy consumption, and the DRL based resource allocation method can maximize the sum capacity of the system.

## 3. System Model and Problem Formulation

We consider concurrent V2I and V2V, based on Mode 3 specified in 3GPP cellular V2X framework, which consists of multiple RRH and multiple vehicle equipment (VE). The RRHs are deployed at the center of intersections and connected to BBU pool through fronthaul link; the VEs are distributed across the network and move on the roads. The RRHs. In Mode 3, the RRHs allocate communication resources to V2I mode VEs, for the subsequent reuse by V2V mode VEs. A V2I mode VE can interfere with V2V mode VE while multiple V2V mode VEs share the same resources. The total VE set can be expressed as $\mathbb{U}$, and consists of a V2I VE set $\mathbb{C}$ and a V2V VE set $\mathbb{V}$. We denote the set of RRHs as $\mathbb{S}$ within the total available bandwidth $W^{total}$. The total bandwidth $W^{total}$ is divided into $\mathbb{K}$ resource block (RB)s, allocated to each V2I VE. The used symbols are summarized in Table 1.

**Table 1.** List of symbols and their description.

| Symbol | Description |
|---|---|
| $\mathbb{U}, \mathbb{C}, \mathbb{V}$ | Set of total VE, V2I VE, V2V VE |
| $\mathbb{S}, \mathbb{K}$ | Set of total RRHs, resource blocks |
| $W^{total}$ | Total available bandwidth |
| $p_{tr}^0, p_k^v$ | Transmission power of RRH, V2V VE $v$ at the $k$th RB |
| $N$ | Noise power spectral density |
| $\gamma_k^c, \gamma_k^v$ | SINR of $c$th V2I and $v$th V2V VE at the $k$th RB |
| $R$ | Total system capacity that means sum achieved data rate of total VE |
| $P$ | Total system energy consumption |
| $P_{\mathbb{S}}$ | Power consumption of total RRHs |
| $P_{fronthaul}$ | Power consumption of total fronthaul links |
| $EE$ | System energy efficiency |
| $\gamma, \gamma_0$ | SINR of VE, SINR constraint |
| $\tau, \tau_{max}$ | Outage probability of system, Outage probability constraint |
| $p_{max}$ | Maximum transmission power of VE |

In V2V communication, it is imperative to ensure stringent latency and reliability requirements when safety-critical messages are being transmitted. However, it is difficult to formulate the latency and reliability requirements as constraints of the optimization problem. Therefore, according to Little's law, the latency requirement is converted into data queue length, and the reliability requirement is converted into outage probability [18]. The outage probability is defined as the probability that the signal to interference plus noise ratio (SINR) of the VEs is lower than the predefined SINR threshold [19]. Increasing the outage probability results in packet loss, and reveals re-transmission occurrence leading to reduced reliability. When reusing cellular resources, there is mutual interference between V2I and V2V communication. Thus, the SINR is derived to consider the interference between V2I and V2V communication. The SINR of $c$th V2I VE at the $k$th RB is

$$\gamma_k^c = \frac{p_{tr}^0 \cdot g_k^{sc}}{N + \sum_{v \in \mathbb{V}} p_k^v \cdot g_k^{cv} + \sum_{s' \in \mathbb{S}, s' \neq s} p_{tr}^0 \cdot g_k^{s'c'}} \tag{1}$$

where $p_{tr}^0$ is the transmission power of RRH; $g^s c_k$ is the channel gain between the V2I VE $c$ and the associated RRH $s$ on $k$th RB. $N$ is the noise power spectral density and $p_k^v$ is the transmission power of V2V VE $v$ at the $k$th RB. The SINR of V2V VE $v$ at the $k$th RB is

$$\gamma_k^v = \frac{p_k^v \cdot g_k^{vv'}}{N + p_k^c \cdot g_k^{cv} + \sum_{j' \in \mathbb{V}, j \neq v} p_k^j \cdot g_k^{jv}} \tag{2}$$

where $p_k^v$ is the transmission power of V2V VE $v$ at the $k$th RB. The total system capacity is

$$R = W^{total} \cdot \sum_{c \in \mathbb{C}} \sum_{k \in \mathbb{K}} \left\{ log_2(1 + \gamma_k^c) + \sum_{v \in \mathbb{V}} log_2(1 + \gamma_k^v) \right\} \tag{3}$$

where $W^{total}$ is the system bandwidth. Only the power consumption of the RRHs and the associated fronthauls were considered. The macro BS is only involved in voice services; hence, its energy consumption is disregarded. The power consumption model of RRH is

$$P_{\mathbb{S}} = \sum_{s=1}^{\mathbb{S}} \left( \pi_{\mathbb{S}} + \Delta slope \sum_{c=1}^{\mathbb{C}} a_c^s p_{tr}^0 \right) \tag{4}$$

where $\pi_{\mathbb{S}}$ is the circuit power of the RRH, $\Delta slope$ is the slope of load-dependent power consumption of RRH, as reported in [20], and $a_c^s$ is the association indicator for V2I VE $c$,

with values of 1 for association and 0 for nonassociation. The power consumption model of the fronthaul links is

$$P_{fronthaul} = \sum_{s=1}^{\mathbb{S}} \left( \pi_{fronthaul} + \varphi \cdot t_s \right) \tag{5}$$

where $\pi_{fronthaul}$ is the circuit power from the fronthaul transceiver and switch, $\varphi$ is the power consumption per bit/s and $t_s$ is the traffic associated with RRH $s$. The power consumption model of the total system is

$$P = P_{\mathbb{S}} + P_{fronthaul} \tag{6}$$

The system energy efficiency can be defined as

$$EE = \frac{R}{P} \tag{7}$$

Our main goal is to maximize the energy efficiency of the system, which can be defined as

$$\max_{\{\mathbb{C}, \mathbb{V}, \mathbb{K}\}} EE, \tag{8}$$

$$
\begin{aligned}
\text{s.t.} \, C1 &: \gamma \geq \gamma_0, \quad \forall c, v, k \\
C2 &: \tau \leq \tau_{max} \\
C3 &: 0 < p_k^v \leq p_{max}, \quad \forall v, k \\
C4 &: \sum_{k \in \mathbb{K}} f_{c,k} \leq 1, f_{c,k} \in \{0, 1\}
\end{aligned}
\tag{9}
$$

where $\gamma$ is the SINR of VE; $\gamma_0$ is the SINR threshold; $p_{max}$ is the maximum transmission power of VE; $\tau$ is the outage probability of the system; and $\tau_{max}$ is the maximum outage probability constraint; $f_{c,k}$ is the allocation indicator with values of 1 for allocation to V2I VE $c$ at $k$th RB and 0 for non-allocation. Of these constraints, C1 ensures that VEs satisfy the SINR constraint; C2 ensures that the outage probability in the system does not exceed the threshold outage probability constraint; C3 ensures that the transmission power of V2V transmitters does not exceed the maximum power level because of the critical interference problem; C4 denotes that each V2I VE can be allocated to an RB. It means that each V2I VE can be allocated to only one RB, and other V2I VE cannot share the allocated the RB.

## 4. Proposed Algorithm

In this section, a vehicle clustering and resource allocation algorithm based on deep Q-learning is introduced. Our algorithm consists of three parts; RRH grouping method, vehicle cluster formation, and deep Q-learning based resource allocation method. First, a method for RRH grouping is introduced to reduce the load caused by deep Q-learning. A technique is also introduced in which vehicles form a cluster to allocate communication resources and transmit periodic messages.

### 4.1. RRH Grouping Method

When the RRH become deep Q-learning agents and perform learning, frequent state updates occur, resulting in critical communication overhead and maintenance cost. To reduce the overhead, similar RRHs are grouped. The similar RRHs means RRHs having similar traffic patterns consisted of amounts of servicing data and vehicles, and interference environments. Moreover, location of RRH is used to consider the similarity of RRHs because the data traffic of vehicular networks has spatial regularities. Then, each RRH group becomes an agent and learns to maximize system energy efficiency. In clustering similar RRHs into a group, we consider four factors; traffic volume, extent of data services, interference of VEs, and location of RRH. Traffic volume stands for the number of VEs going through each RRH per time unit. The extent of data services refers to the resources

the RRH serves VEs per hour. The interference of VEs means the average of the SINR values of VEs serviced by RRH per time unit. Considering the aforementioned factors, similarities of neighboring RRHs are calculated and stored in an array for each RRH. In addition, $N_s$ RRHs are selected and assigned to the center of the cluster; those with maximum similarity are further incorporated into clusters for each RRH. The minimax method, which can minimize possible losses in game theory, is adopted as a method of selecting the center.

### 4.2. Vehicle Cluster Formation

In V2X communication, *SidelikeUEInformation* message to request resource allocation and *MeasurementReport* message to report location information use the same communication channel [8]. Therefore, in environments where there are many VEs, the RRHs are under heavy traffic load because of periodic messages. To solve this problem, multiple VEs form one cluster, and a cluster head (CH) receives a message from each cluster member (CM) and configures the messages into a jumbo message. A message, called a single frame, consisting of a header and payload, is transmitted periodically between VEs to examine the surrounding environment for ID, time, location, velocity, and so on. CH creates a jumbo frame with multiple single frames consisting of one header and one large payload. Because the headers of multiple single frames become one jumbo frame header, the message size and communication overhead decrease. When a single frame is transmitted at every time unit $t$, the traffic associated with the single frames is

$$\sum_{t \in \mathbb{T}} \left( l_{header} + l_{payload}^t \right) \cdot N_u^t \tag{10}$$

where $l_{header}$ is the length of the header; $l_{payload}^t$ is the length of the payload; and $N_u^t$ is the number of VEs at time $t$. When CH configures a jumbo frame with multiple single frames, the traffic of the jumbo frames is

$$\sum_{t \in \mathbb{T}} \left( l_{header} + l_{payload}^t \cdot N_{CM}^t \right) \tag{11}$$

where $N_{CM}^t$ is the number of CM. The cluster formation step consists of as followings.

1. First step is initializing step, and it occurs when no cluster is configured or the initial step of algorithm. If the initial step, all the VE associate with RRH serving the highest SINR and are set V2I mode. In first step, V2I mode VE with the highest value $F_C H^u$ is selected as first CH and $F_{CH}^u$ are calculated as follows:

$$F_{CH}^u = \gamma_k^c \cdot PPPP^u \tag{12}$$

where $PPPP^u$ is the prose-per-packet-priority (PPPP) value of VE $u$. PPPP is the feature of priority in V2X data transmission. Frames to which different PPPP values are assigned wait for transmission of messages from different VEs, where the packets are ordered with PPPP values between 0 and 1 [21]. This ensures that the CH is the first to be placed in the jumbo frame during configuration of the frame to satisfy the PPPP. That is, a payload of CH is located at the first of the jumbo frame. Next, payloads are sequentially arranged according to the $F_{CH}^u$, and CM following CH becomes the next CH.

2. Next step is cluster formation and deformation step, and it occurs in the following situations: (i) When the VE moves to another cell; (ii) when the payload of the jumbo frame exceeds the capacity of the acceptable payload; (iii) when the communication distance between CM and CH exceeds the V2V communication distance constraint. In cluster formation and deformation step, the neighboring vehicles decide whether associating with the cluster or not through comparing receiving SINR between serving RRH and adjacent CH. In this process, the SINR of the serving RRH is compared for each vehicle with those of the neighboring CHs using the messages that each vehicle

broadcasts consisting of ID, location, and mobility speed at every time unit. If the SINR of one CH is higher than that of another, then that CH will be clustered with the CH providing the highest SINR.

3. The last step occurs when VEs cannot join any cluster or receive higher SINR from RRH than the SINR from neighboring CHs. Then, the VEs that do not belong to the cluster are set to the V2I mode.

However, if the same vehicle becomes CH in succession, an additional load, such as constructing jumbo frames as CH, will be added. In other words, maintaining the CH status for the same vehicle would incur significant overhead and therefore would have to be handed over to other vehicles at certain time intervals. The CH designates the VE, following the current CH in the jumbo frame, as the next CH. The jumbo frame is stacked in queue in the order of the messages received from CM, as described in Figure 1. Hence, no additional load is created for CH. However, a jumbo frame has a sub-optimal relationship with the optimal CH of the cluster.

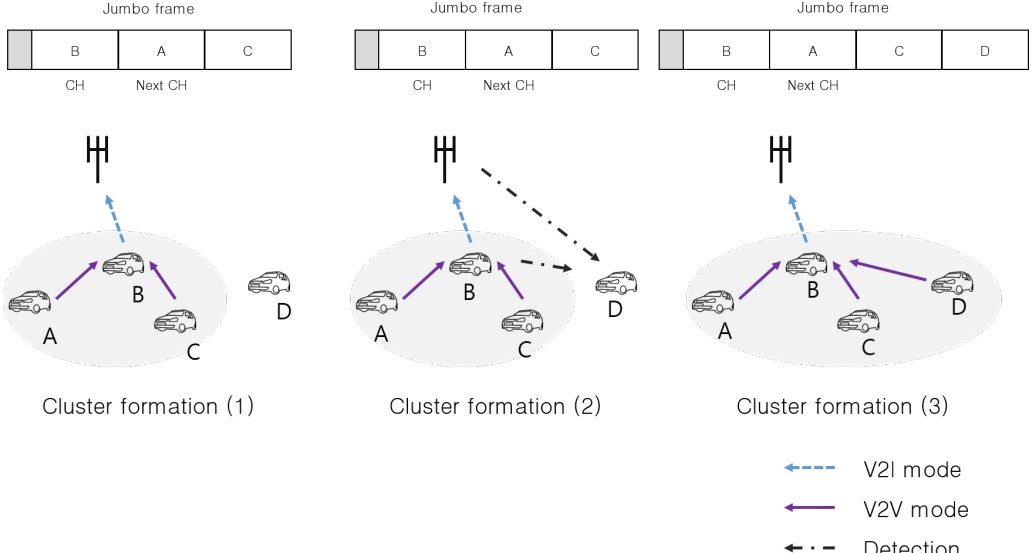

**Figure 1.** Cluster formation.

Once the size of a cluster is fixed, a larger cluster size can reduce the V2I-mode VEs. A larger cluster can also increase the data rate and reduce interference based on the decrease in the number of V2I-mode VEs. However, there is additional overhead depending on the number of messages to be processed while maintaining the CH of large clusters. If the cluster size is small, the number of V2I-mode VEs will increase and the RRH load will also increase. In addition, fixed-size clusters with low vehicle density may increase the distance between CMs, resulting in poor QoS. Therefore, a flexible cluster size is required.

*4.3. Deep Q-Learning Based Resource Allocation Method*

The sharing of communication resource among V2V VEs, can cause high complexity in interference controls, continuous-value state, and large action space. Therefore, the resource allocation problem was formulated as a Markov decision process. MDP can be defined by a tuple $(S, A, p, r)$ where $S$ is a set of states; $A$ is a set of actions; $p$ is a transition probability from state $s$ to $s'(s, s' \in \mathbb{S})$; and $r$ is the immediate reward. At each time $t$, the V2I mode VE makes a request to allocate communication resources to the serving RRH. RRH observes the state, $s_t$, and selects an action $a_t$, accordingly. After the action is executed, the state $s_t$ of the environment changes to a new state $s'_t$ and the agent receives a reward $r_t$. In DQN, updates of Q-table are converted into the updates of network weights. The experience replay and fixed target network are developed to accelerate training process, so

it can improve the convergence. In fixed target network, the update of the target Q-network is adopted to accelerated the training process. Figure 2 is illustrated to describe DQN.

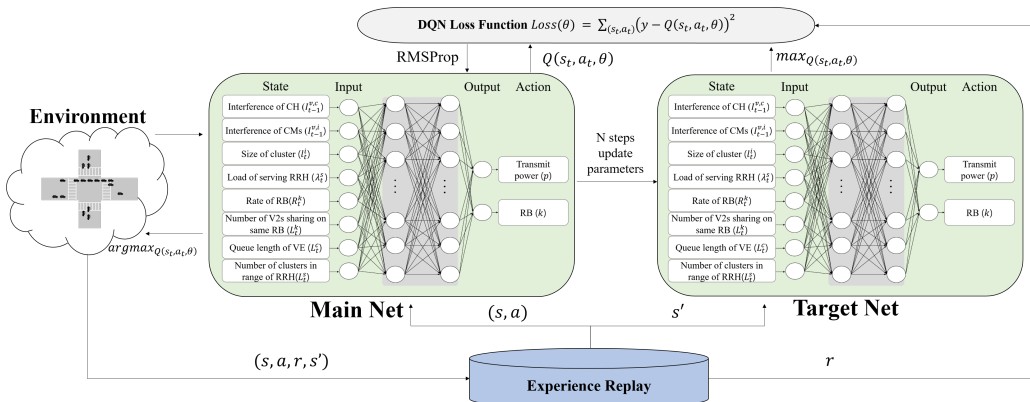

**Figure 2.** The structure of DQN.

The details of the tuples in our proposed method are defined as follows:

1. The state of V2I VE $c$, the observed state $s_t$ at time unit $t$, consists of eight pieces of information: The interference received from CH $I_{t-1}^{v,c}$; the interference received from the CMs in vehicle cluster $i$ $I_{t-1}^{v,i}$; the size of cluster $l_t^i$; the load of serving RRH $\lambda_t^s$; the rate of RB $k$ usage $R_t^k$; the number of VEs sharing the same RB $k$ $L_t^k$; the queue length of VE $c$ $L_t^c$; and the number of clusters in range of RRH $s$ $L_t^s$. Thus, the state can be described as follows:

$$s_t = \{I_{t-1}^{v,c}, I_{t-1}^{v,i}, l_t^i, \lambda_t^s, R_t^k, L_t^k, L_t^c, L_t^s\} \tag{13}$$

2. The action of each V2I VE $c$ is defined as

$$a_t = \{p, k\} \tag{14}$$

where $p$ is the transmit power level, which is divided into discrete intervals, $[0, p_{max}]$ and $k$ is RB.

3. To maximize the system capacity and guarantee the QoS requirements, the reward is defined as

$$r_t = \begin{cases} \omega \cdot \gamma_{k_i} + (1 - \omega) \cdot (T_0 - T_{t-1}), & \sigma_{t-1} \leq \sigma_0 \\ -1, & \sigma_t > \sigma_0 \end{cases} \tag{15}$$

where $T_0$ is the maximum latency constraint; $T_{t-1}$ is the latency at time $t-1$; $T_0 - T_{t-1}$ represents the latency penalty; $\sigma_{t-1}$ is the outage probability at time $t-1$.

For optimal policy, the deep Q-network is trained to approximate the Q-function. The deep Q-learning agents deploy each RRH group, and the optimal policy is determined based on the following update equation.

$$\begin{aligned} Q_{t+1}(s_t, a_t) = {} & Q_t(s_t, a_t) \\ & + \alpha \left[ r_{t+1} + \beta \cdot \max_a Q_t(s_{t+1}, a_{t+1}) - Q_t(s_t, a_t) \right] \end{aligned} \tag{16}$$

where $\alpha$ is the learning rate; $\beta$ is the discount factor [22]. The optimal policy $\pi^*$ for state $s_t$ can be expressed as

$$\pi^*(s_t) = \max_{a_t} Q_{t+1}(s_t, a_t) \tag{17}$$

The deep Q-network improves the Q-learning by combining DNN with Q-learning, to approximate the Q-function by a DNN. The deep Q-learning updates the Q-network with

weight $\{\theta\}$ [14]. When $\{\theta\}$ is determined, $Q(s_t, a_t)$ becomes the output of the DNN. The Q-network updates the weight $\theta$ to minimize the following loss function.

$$Loss(\theta) = \sum_{(s_t, a_t)} (y - Q(s_t, a_t, \theta))^2 \tag{18}$$

where

$$y = r_t + \max_{a \in A} Q(s_t, a_t, \theta) \tag{19}$$

where $r_t$ is the corresponding reward.

## 5. Simulation Results and Discussions

In this section, the performance of the proposed algorithm was compared with that of other existing algorithms. Simulations were carried out using Tensorflow 2.0 and the Simulation of Urban Mobility (SUMO) simulator [23]. In the simulations, the multi-cell H-CRAN environment was considered; the parameters are summarized in Table 2. The parameters and system requirements are set in accordance with the 3rd Generation Partnership Project (3GPP) specifications release 15 and 16 [20,24]. In addition, Rayleigh fading, log-normal shadowing, and the path-loss model($140.7 + 36.7 \log(distance)$) were considered [25]. The DQN adopted for the simulation was a fully connected neural network consisting of an input layer, a hidden layer, and an output layer. The number of neurons in the hidden layer was 256 and ReLu was utilized for the activation function. The DQN parameters had learning rate $\alpha = 0.01$, discount factor $\beta = 0.9$. The replay memory size is 3000, network update frequency is 2, and target network update frequency is 30. Each network update occurs every unit episodes.The latency and reliability requirements were 10 ms and an outage threshold of 3 dB, respectively.

**Table 2.** Parameters used in the simulation.

| Parameter | Notation | Value |
|---|---|---|
| Noise power spectral density | $N$ | $-174$ dBm/Hz |
| Total bandwidth | $W^{total}$ | 100 MHz |
| SINR threshold | $\gamma_0$ | 0.5 dBm |
| Maximum outage probability constraint | $\tau_{max}$ | 0.05 |
| Circuit power of RRH | $\pi_{\mathbb{S}}$ | 4.3 W |
| Slope of RRH | $\Delta slope$ | 4.0 |
| Circuit power of fronthaul transceiver and switch | $\pi_{fronthaul}$ | 13 W |
| Power consumption per bit/s | $\varphi$ | 0.83 W |
| Transmission power of V2I mode | $p_k^c$ | 23 dBm |
| Transmission power of RRH | $p_{tr}^0$ | 24 dBm |
| Cluster size of RRH grouping | $N_s$ | 5 |

In the multi-cell environment, 43 RRHs were deployed at the center of intersections, and the vehicles were distributed and moved on the roads. The Luxembourg SUMO traffic Scenario [26] dataset was used for the mobility and intersection simulations to create a scenario that would meet common requirements and contain realistic traffic demand and mobility patterns. With this dataset, we composed two types of scenarios: Low traffic and high traffic. As the Figure 3, the high traffic scenario shows mobility during rush hour peaks around 20:00 with high traffic demands for one day. In the low traffic scenario, the required traffic demand is about half that of the high traffic scenario.

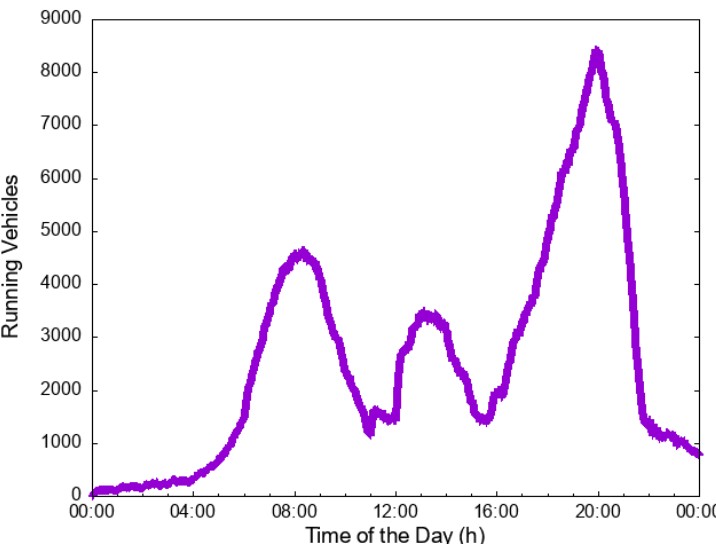

**Figure 3.** Traffic of day.

To evaluate the performance of the proposed algorithm, we used four metrics: Average SINR, average achievable data rate, system energy efficiency, and outage probability. The SINR is the important factor to show interference condition of vehicles, so the average SINR of total vehicles is used to show the increase or decrease in interference due to resource allocation. The average achievable data rate means system capacity that can be achieved in a given bandwidth, and it used to show efficiency of resource allocation method. The system energy efficiency means the achievable data rate per energy consumption, it is the main objective of our proposed algorithm. So the system energy efficiency is used to evaluate main performance of proposed algorithm. The outage probability is also used to show ensuring QoS constraint. We also simulated with four metrics as a function of the variation in V2V communication distance and maximum cluster size. Because V2V communication distance is a parameter related to cluster size and QoS of the vehicle, we compared the performance as a function of the change in V2V communication distance.

In the evaluation of the performance of the proposed algorithm, the other three algorithms used were as follows: A baseline algorithm, clustering with DQN based mode selection and resource allocation algorithm, and DQN based resource allocation algorithm. First, we used the SINR-based algorithm in which VEs communicate with only the V2I mode. In this algorithm, VEs associated with RRHs that served the highest SINR. It was denoted as "Method with SINR-based V2I communication (*BA*)". Second, for vehicle clustering comparison, the DQN based mode selection and resource allocation algorithm in [17] was used to learn to maximize the system achievable data rate with vehicle clustering, mode selection and resource allocation method. We denote this as "Method with clustering and DQN (*Compare1*)". Unlike the proposed algorithm, the communication mode and resource allocation of the vehicle were determined in consideration of the channel state of the individual vehicle in *Compare1*. Therefore, this algorithm was used to compare performances with the proposed algorithm which is considering the sharing state of resources in use with the vehicle cluster. Third, for a comparison with the DQN based resource allocation algorithm, the algorithm in [16] was used to learn to maximize the system achievable data rate with the resource allocation method. We denote this as "Method with DQN (*Compare2*)". Unlike the proposed algorithm, *Compare2* only considered channels and resource states that can be checked at RRHs. Therefore, this algorithm was used to compare performance with the proposed algorithm that considered local information of vehicles.

The average SINR and average achievable data rate for two traffic load scenarios are compared as a function of the V2V communication distance, as displayed in Figures 4 and 5. In Figures 4 and 5, as the V2V communication distance increases, the average SINR and achievable data rate decrease because of the increasing distance between V2V VEs. The

proposed algorithm performs better than the other algorithms. When compared with *Compare1*, the clustering algorithm *Compare1* has a higher ratio of V2I mode VEs and higher interference, resulting in lower average SINR. The achievable data rate also decreases due to increasing interference. This is because the proposed algorithm selects the VE with the highest SINR as CH and enters the cluster with the highest SINR. Compare1 exhibits lower SINR and achievable data rate than *Compare2*, which does not employ a clustering method. *Compare2* selects a communication mode in which VEs can receive the best SINR. Because the proposed algorithm allocates resources in units of vehicle clusters, the average SINR and achievable data rate are higher than those of *Compare2* due to low interference. Compared to the low traffic scenarios, the performance of high traffic scenarios is observed to be better. This is because the vehicle density in high traffic scenarios is higher than the density in low traffic scenarios making V2V communication possible with VEs relatively closer to each other.

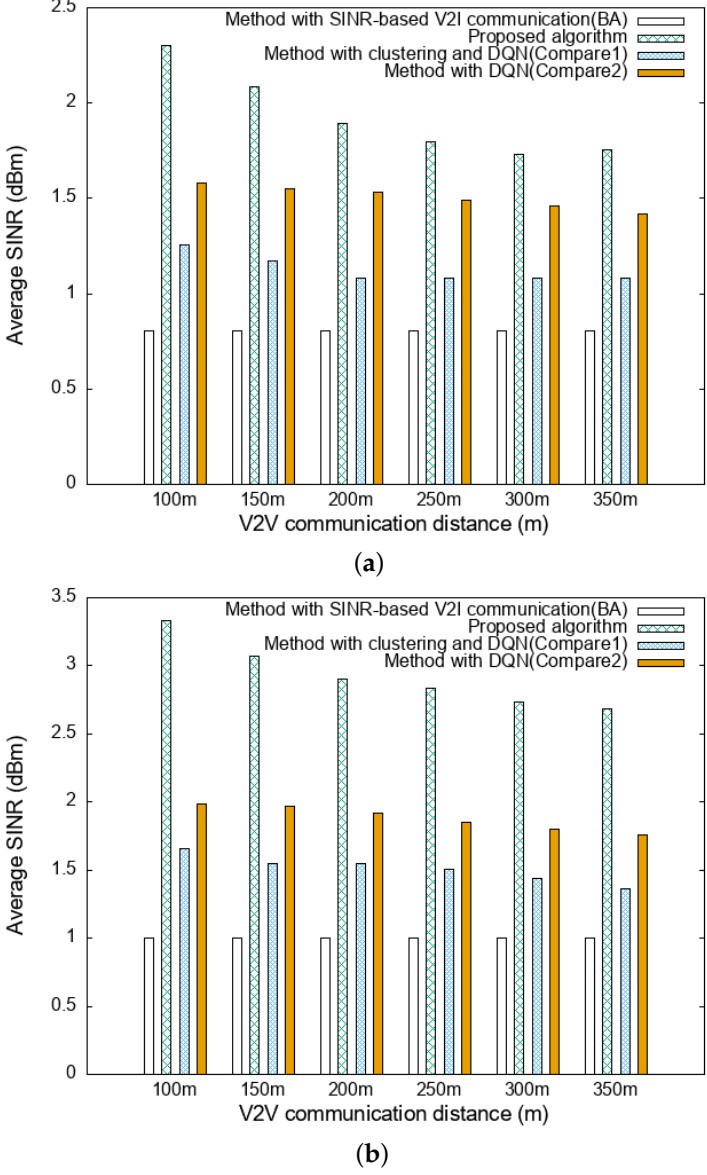

**Figure 4.** Comparison of average SINRs with various V2V distance: (**a**) With low scenario; (**b**) with high scenario.

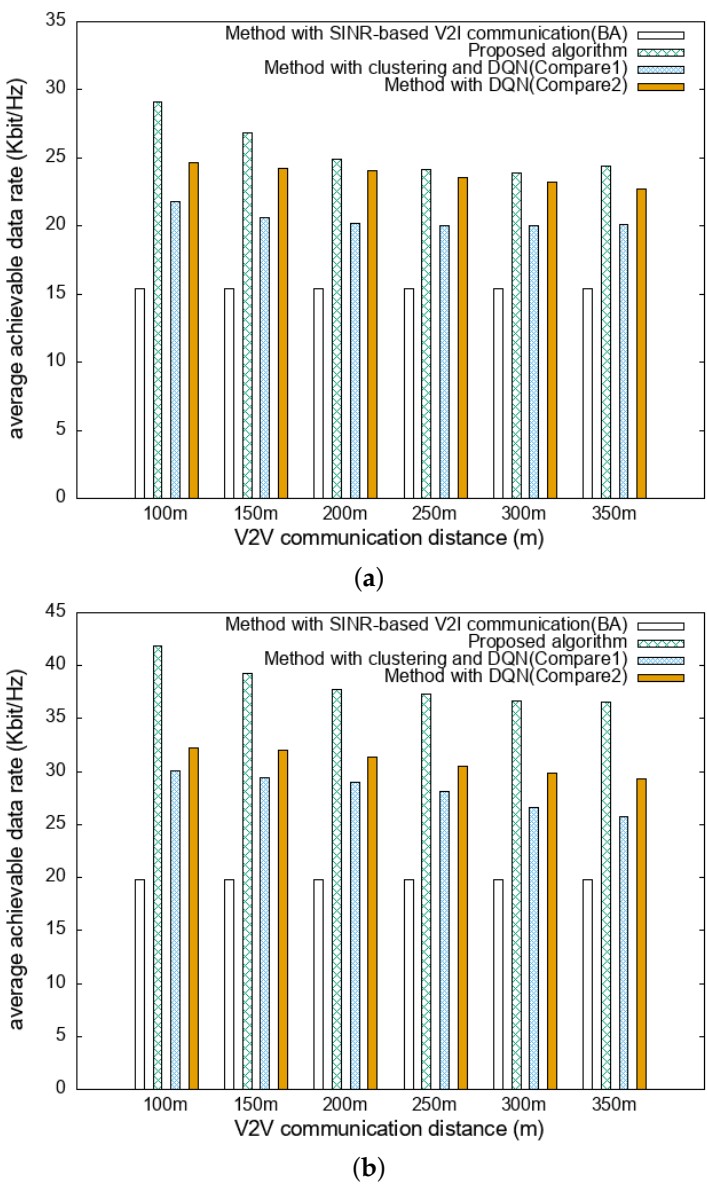

**Figure 5.** Comparison of achievable data rates with various V2V distance: (**a**) With low scenario; (**b**) with high scenario.

The outage probability for the different traffic scenarios are compared as a function of the V2V communication distance, as shown in Figure 6. The comparisons included only three algorithms: The proposed algorithm, *Compare1*, and *Compare2*. *BA* was excluded because of its low outage probability as it only communicates with V2I through SINR-based user association. In the case of the proposed algorithm and *Compare1*, both methods employed clustering because the V2V communication ratio was high. The outage probability of the proposed method was lower than that of *Compare2* however, due to low interference. When the V2V communication distance is increased, CH is capable of receiving optimal SINR, thus minimizing the interference. In Figure 6 it is observed that the outage probability is lower for the high scenario case than for the low scenario case. This is because in the high traffic scenario, the vehicles in the environment are more likely to be relatively close to the communication target because of the higher density of VE. The outage probability of the proposed algorithm is higher than that of *Compare1* because there are more V2V mode VEs in the proposed algorithm than in *Compare1*, and there are more VEs that have the high variance of SINR. Within 150 m and 200 m, the proposed algorithm has lower outage probability than *Compare2* because the V2V communication

rate is reduced as a result of high interference from the short V2V communication distance in high traffic scenarios.

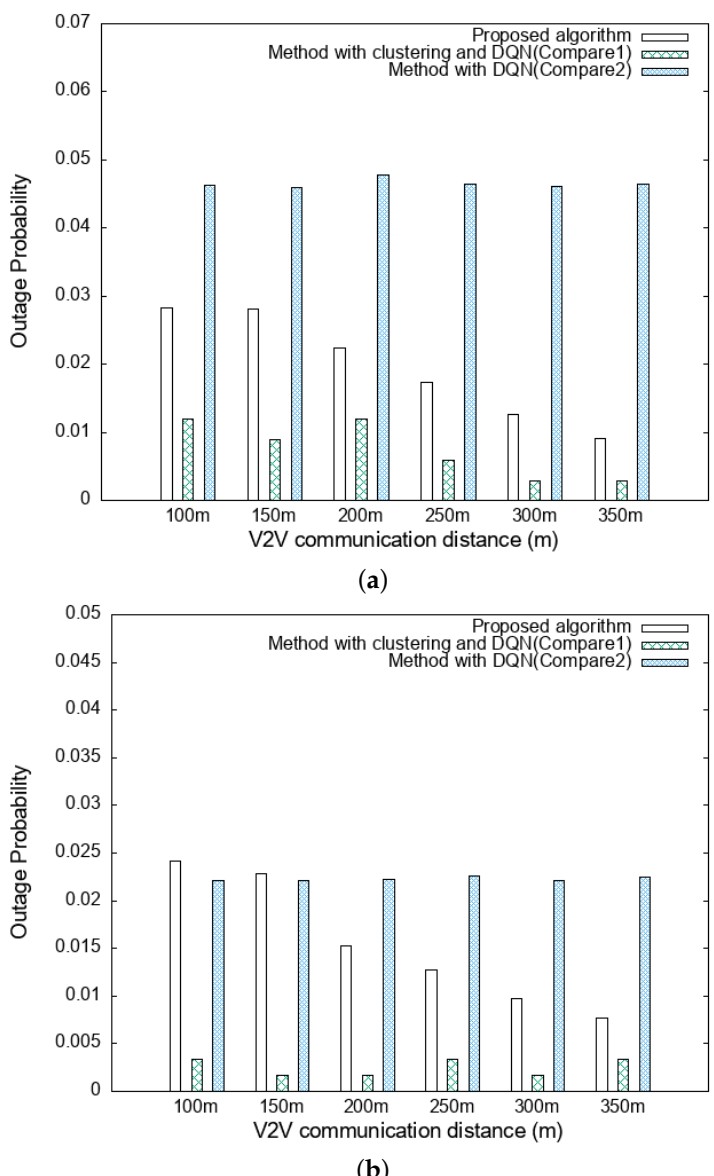

**Figure 6.** Comparison of outage probabilities with various V2V distances: (**a**) With low scenario; (**b**) with high scenario.

The system energy efficiency for different traffic scenarios are compared as a function of the V2V communication distance, as displayed in Figure 7. It is observed in Figure 7 that the system energy efficiency of all algorithms increases as the V2V communication distance increases. This is because when the V2V communication distance increases, the cluster size can increase to the extent that it is close to the limit. That is, as the number of V2V mode VEs increases, system energy consumption of the RRHs decreases and system energy efficiency increases. In the high traffic scenario, the performances of the proposed algorithm and that of *Compare1* are particularly improved because the vehicle density is higher than in the case of the low scenario, allowing more VEs to communicate in the V2V mode, which could reduce system energy consumption. In addition, the proposed algorithm displays the highest energy efficiency by applying the clustering method to reduce system energy consumption and a DQN-based resource allocation to increase the achievable data rate.

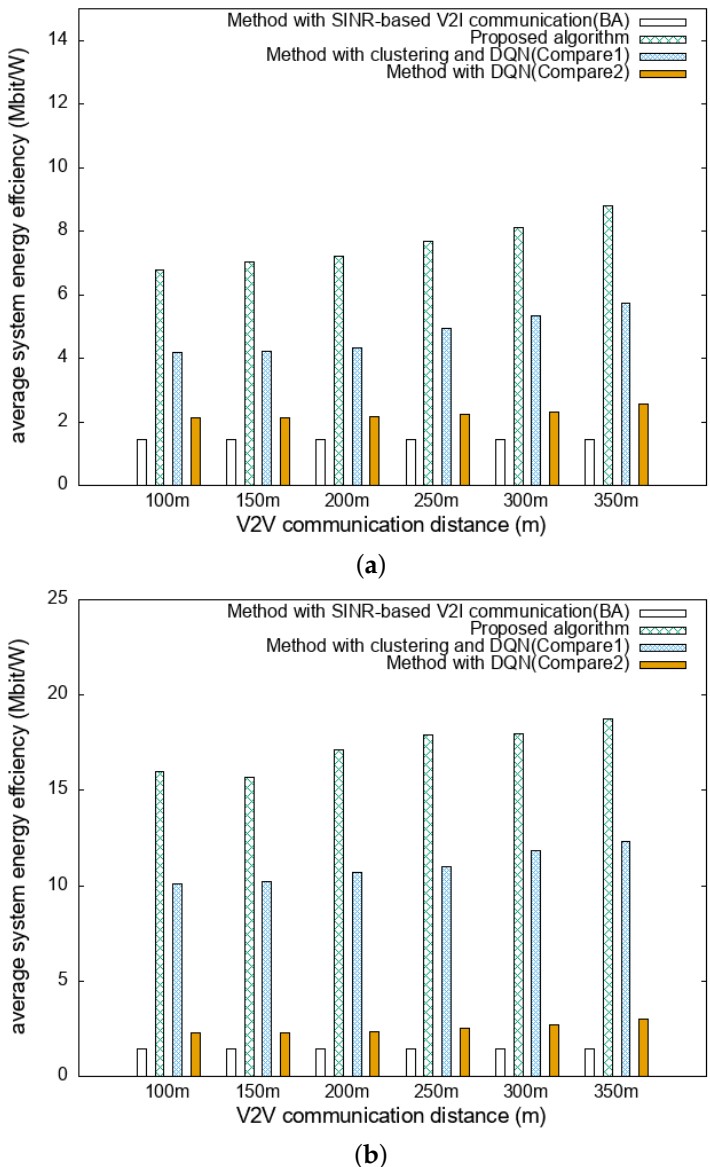

**Figure 7.** Comparison of system energy efficiencies with various V2V distances: (**a**) With low scenario; (**b**) with high scenario.

In Figures 8–10, only the proposed and *Compare1* algorithms are compared as a function of vehicle density change to evaluate the performance of the clustering method. The vehicle density change denotes a change in the average distance between vehicles when the total number of vehicles is fixed, which is different for the low and high scenarios in which the number of vehicles is changed. Unlike low and high scenarios, only the average distance between vehicles changes with interference based on the distance from the communication target.

The average SINR and average achievable data rate for difference traffic scenarios are compared as a function of the cluster size, in Figures 8 and 9. The cluster size indicates size limit, and VEs can only be clustered up to the cluster size. In Figures 8 and 9, when the density is the same, the proposed algorithm displays the highest performance, and *Compare1* displays the next highest performance. This is because the VEs that can receive the highest SINR in the proposed algorithm are selected as CH to configure the clusters. It can be also observed that performances in the case of high traffic scenarios are better than those of the low scenarios because the number of VEs in the high scenario increases as the distance between the vehicles is decreased. Interference also increases, but the performance

is still better because the benefits of getting closer to the communication target outweigh other considerations. The increase in density is also a result of the decrease in the average distance between vehicles, increasing the number of clustered vehicles. In each algorithm, the distance between VEs is on average 0.5, which is less than a density of 1. Therefore, as the distance to the communication target becomes closer, the average SINR that is received would increase and thus, the achievable data rate would also increase.

The system energy efficiency for different traffic scenarios is compared as a function of the V2V communication distance, as displayed in Figure 10. In Figure 10, it is observed that the system energy efficiency is higher for the high scenario. This is because in the high scenario, the proportion of clustered VEs increases, such that the number of VEs communicating through the RRH decreases. Thus, the system energy efficiency also increases. In addition, it is observed that the system energy efficiency of the proposed algorithm is higher compared to *Compare1* for the same density although *Compare1* is also based on the vehicle clustering method. The number of V2I mode VEs in the proposed algorithm is smaller than that of *Compare1*. As a result, system energy consumption is relatively small and interference is reduced, resulting in a higher achievable data rate.

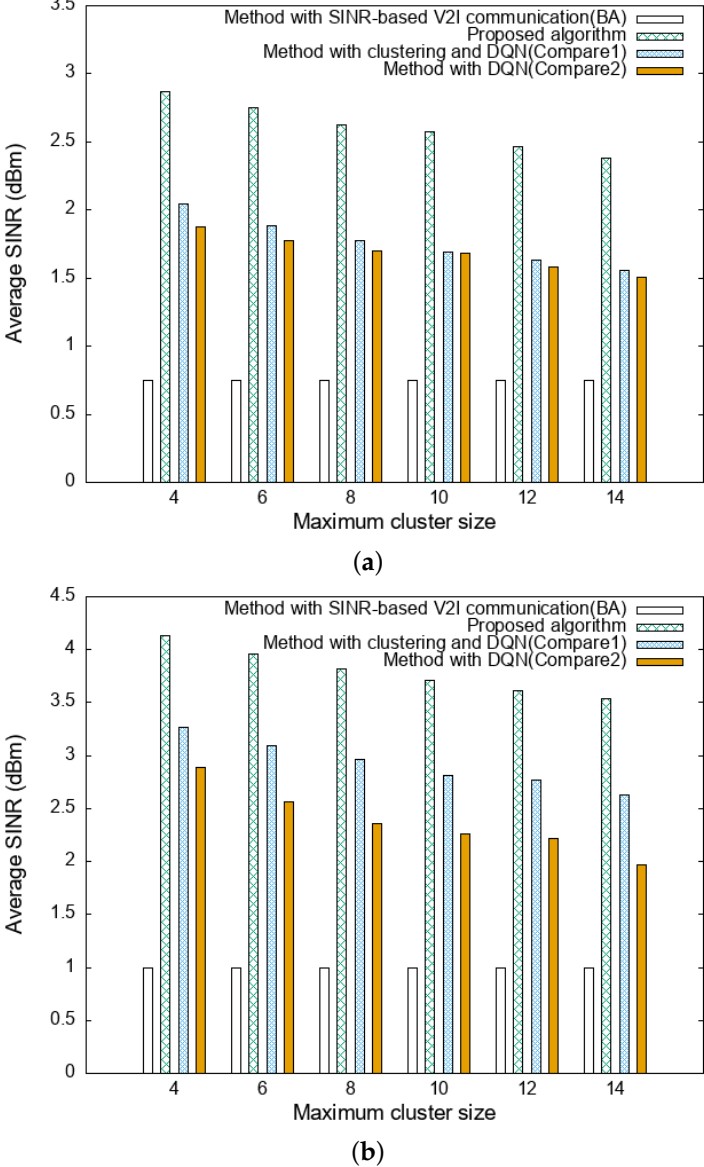

**Figure 8.** Comparison of average SINRs with various maximum cluster sizes: (**a**) With low scenario; (**b**) with high scenario.

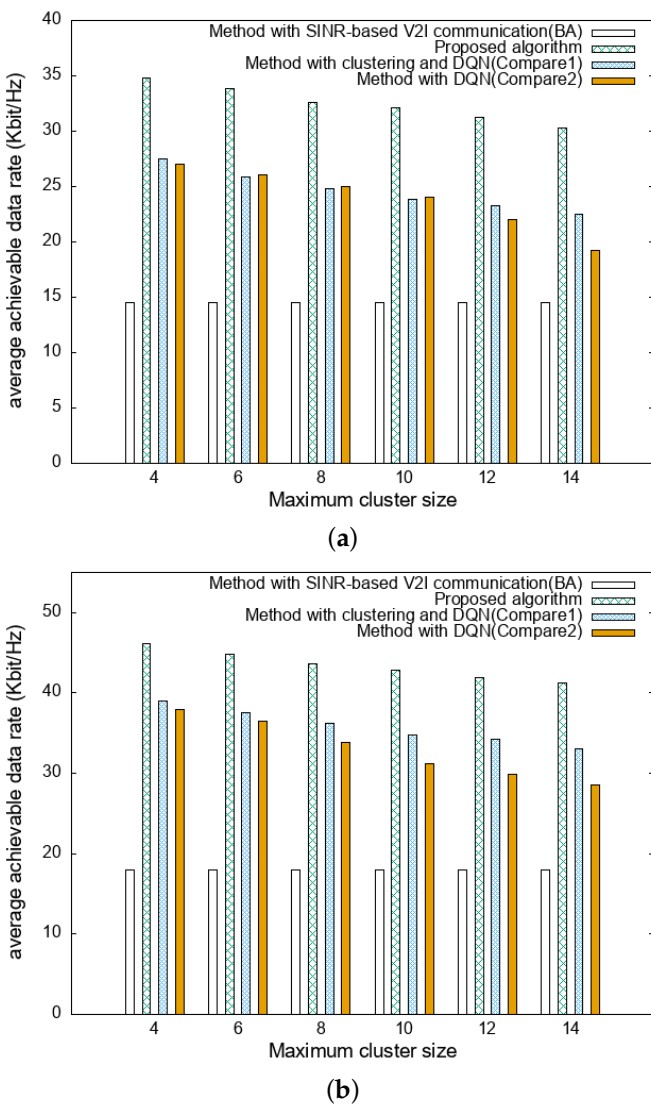

**Figure 9.** Comparison of achievable data rates with various maximum cluster sizes: (**a**) With low scenario; (**b**) with high scenario.

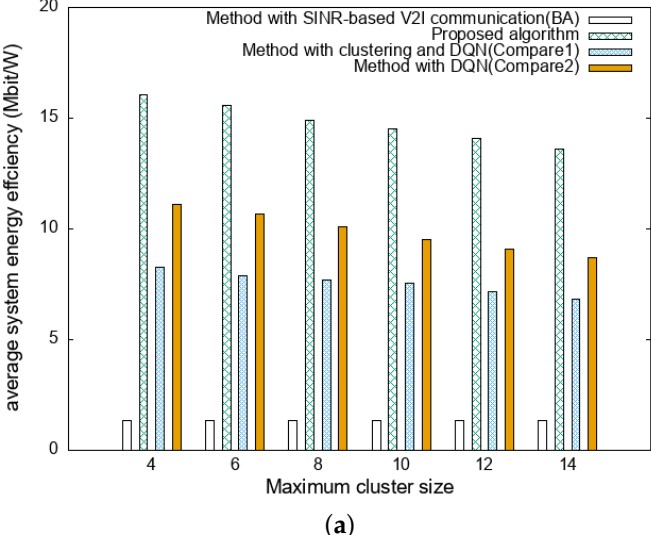

**Figure 10.** *Cont.*

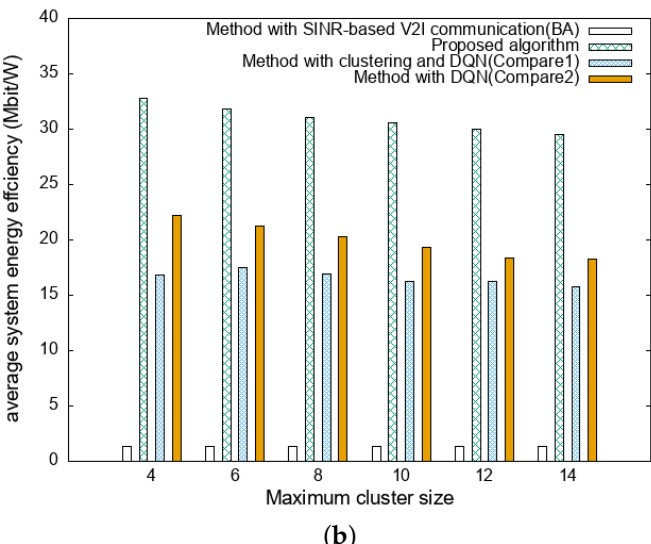

**Figure 10.** Comparison of system energy efficiencies with various maximum cluster sizes: (**a**) With low scenario; (**b**) with high scenario.

The proposed algorithm exhibited the best performance in terms of system energy efficiency, achievable data rate, and average SINR. The proposed algorithm maximized energy efficiency while ensuring QoS. As mentioned earlier, the system energy efficiency and achievable data rate have a trade-off relationship. The proposed algorithm used vehicle clustering method and DQN based resource allocation method to achieve the highest performance compared to other algorithms.

## 6. Conclusions

A DRL based resource allocation algorithm with a clustering method was proposed in this work for cellular V2X communication. The maximization problem of system energy efficiency was formulated using the Markov decision process. The RRH grouping and vehicle clustering method was designed to minimize signaling overhead and communication complexity. A DRL based resource allocation method was adopted to maximize system capacity. To demonstrate that the proposed algorithm outperformed the other algorithms, simulations were carried out with a SUMO simulator to compare system energy efficiency, achievable data rates, average SINR and outage probabilities.

In real-world vehicular networks, it is difficult to collect channel state information from each vehicle in the network. The training data are also distributed among vehicles, causing bandwidth overhead from data upload. Federated learning has the advantage of distributed learning [27,28]. Further studies should be performed that include federated learning to solve the aforementioned problems, to relieve the load of fronthaul links and RRHs, and to manage the resources efficiently in the presence of rapidly changing mobility.

**Author Contributions:** Conceptualization, H.P. and Y.L.; Methodology, H.P.; Software, H.P.; Writing—Review & Editing, H.P. and Y.L.; Supervision. Y.L. All authors have read and agreed to the published version of the manuscript.

**Funding:** This work was supported by the National Research Foundation of Korea(NRF) grant funded by the Korea government (MSIT) (No.2021R1F1A1047113).

**Conflicts of Interest:** The authors declare no conflict of interest.

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
