# Peer review of "Deep Reinforcement Learning Based Resource Allocation with Radio Remote Head Grouping and Vehicle Clustering in 5G Vehicular Networks"

_electronics, doi:10.3390/electronics10233015_

Round 1

Reviewer 1 Report

Simply stating, authors present a mechanism to optimize the communications between vehicles in 5G vehicular networks. The proposal relies on Deep Reinforcement Learning  to setup clusters of vehicles and to perform resource allocation.

The paper is generally well written, but can be improved. In this regard the following suggestions are made:

  1. Section 3, line 142 There is an incomplete sentence "The RRHs."
  2. Section 3, page 4 (or other) introduce a table to summarise the meaning of the diverse terms (V, C, U, K, etc).
  3. Section 3, line 151. Introduce a reference to the Little's law.
  4. Section 3, Eq. (8) and Eq. (9) the term fc,k is not explained in the text, thus not possible to  understand such constraint in the optimisation problem.
  5. Section 4.1 line 193, clarify what are similar RRHs.
  6. Section 4.1, line 193, refer here that the size of the cluster is set according a variable (later on described). If Ntu is  in the table as suggestion 2, it would facilitate the understanding.
  7. Section 4.2 The description of the cluster formation is done in text. Having the same represented as an algorithm with the diverse steps, it would facilitate the process.
  8. All paper, including section 4.2 assure that you reference the images in the text. For instance, Figure 1 is not referenced in the text.
  9. Section 4.3, again authors describe the MDP in the text, why not using a figure to illustrate the states, their transition.
  10. Section 4.3, correct the parenthesis in line 269, it should be [ ... ]
  11. Section 5, can be better described. Atuhors mix methodology with the presentation of results, which leads to some confusion. Describe evaluation methodology, for instance, first the metrics evaluating the results (average SINR, outage probability, ...) should be described earlier. The different algorithms/approaches should also be described and properly labeled to facilitate analysis of results in the graphics.
  12. References, uniformize their use. Some have DOI others not. 

Author Response

Dear Reviewer,

First of all, thank you for your thorough review and comments for our manuscript electronics-1468723. We also thank you for giving us the opportunity to revise and resubmit the manuscript. We have made a major revision to the previous manuscript according to the reviewer’s comments. Comments from reviewers were very helpful in improving the quality of the manuscript. Please note that the changed or added sentences are marked by blue and italic characters in the revised manuscript. A list of specific changes made is attached below. Please let us know if you have further concerns.

Sincerely yours,

Hyebin park, and Yujin Lim

Reviewer 2 Report

This article proposes a solution to reduce the transmission interference of the Internet of vehicles. Specifically, a DRL-based resource allocation method is proposed with RRH grouping and vehicle clustering to maximize system energy efficiency.

Overall the paper is fine, but the following points require improvements.

  1. The draft lacks a detailed description of the DQN training process and the timing of parameter updates.
  2. It is not clear whether the car should first consider joining a cluster or establishing a V2I connection with the RRH when entering a cell.

Author Response

(The authors gave the same response as above.)

Round 2

Reviewer 2 Report

The authors have addressed my comments.